# The Role of Th17-Related Cytokines in Atopic Dermatitis

**DOI:** 10.3390/ijms21041314

**Published:** 2020-02-15

**Authors:** Makoto Sugaya

**Affiliations:** Department of Dermatology, International University of Health and Welfare, Ichikawa Hospital, 6-1-14, Kounodai, Ichikawa, Chiba 272-0827, Japan; sugayamder@iuhw.ac.jp; Tel.: +81-47-375-1111; Fax: +81-47-373-4921

**Keywords:** IL-22, IL-26, subtypes of atopic dermatitis, ustekinumab, wound healing, antimicrobial peptides

## Abstract

T helper-17 (Th17) cells, which mainly produce IL-17, are associated with development of various autoimmune diseases such as rheumatoid arthritis, inflammatory bowel diseases, multiple sclerosis, and psoriasis. IL-17 and related cytokines are therapeutic targets of these diseases. In atopic dermatitis (AD), Th2 cytokines such as IL-4 and IL-13 are regarded to be the main player of the disease; however, Th17 cytokines are also expressed in AD skin lesions. Expression of IL-22 rather than IL-17 is predominant in AD skin, which is contrary to cytokine expression in psoriasis skin. Relatively low IL-17 expression in AD skin can induce relatively low antimicrobial peptide expression, which may be a reason why bacterial infection is frequently seen in AD patients. Failure of clinical trials for investigating the efficacy of anti-IL-12/23 p40 in AD has suggested that IL-17 expressed in skin lesions should not be the main player but a bystander responding to barrier dysfunction.

## 1. Introduction

Atopic dermatitis (AD) is a chronic skin disease characterized by relapsing eczema with pruritus as a primary lesion. Most patients with AD have atopic diathesis such as family history of allergic diseases or past history of asthma, rhinitis, or conjunctivitis [1]. In Japan, AD is not a rare skin disease, whose frequency is about 10–15% until the age of forty [2]. Age-related remission is achieved in a certain proportion of patients with AD, especially when the onset is younger or the symptoms are milder [3,4]. Although some patients develop AD in their adulthood, onset of AD in elderly people is very rare. Other skin diseases such as cutaneous T-cell lymphoma (CTCL), drug eruption, and bullous diseases should be considered as a possible diagnosis. Most of all, CTCL should not be misdiagnosed with AD because immunosuppressive drugs such as cyclosporine and anti-tumor necrosis factor (TNF)-α antibody can exacerbate disease activity of CTCL [5,6,7]. When the diagnosis is not clear, topical steroid or ultraviolet phototherapy should be applied.

AD has significant negative social and economic impacts, substantially decreasing the quality of life of the patients and their families [8,9]. Skin conditions of AD children influence quality of life of both children and their caregivers. Pruritus, emotional distress, and sleep disturbance are big problems for them, which should be paid more attention to. It is important to induce long-term remission. Remission can be achieved by conventional treatment for mild or moderate AD, while it is very difficult for severe AD patients [3,4]. Treatment strategies consist of drug therapy with topical agents such as steroid or tacrolimus, skin care with moisturizers for dry skin, and investigation and elimination of exacerbating factors [1]. The pathogenesis of AD can be explained from the perspectives of the skin barrier, allergic inflammation, and neuroendocrine dysfunction represented by severe pruritus [10]. With regards to allergic immune responses, cytokines expressed by T helper (Th) 2 cells and group 2 innate lymphoid cells such as interleukin (IL)-4, IL-5, and IL-13 have been assumed as main players of AD [11,12,13]. It was also reported that Th1 cytokines like interferon (IFN)-γ were expressed in AD lesional skin [14,15]. In recent years, Th17 cytokines such as IL-17 and IL-22 were reported to be expressed in AD lesional skin [16,17,18,19,20]. In this review, the importance of Th17-related cytokines in pathogenesis and treatment of AD will be discussed.

## 2. Th17 Cells in Pathogenesis of Psoriasis

CD4+ helper T cells, upon activation, develop into different T helper cell subsets. Th17 cells, which are different from Th1 or Th2 cells, produce IL-17, IL-17F, and IL-22 as well as IL-21 [21]. Th17 cells are essential in clearing pathogens during host defense reactions. IL-17 and IL-22 are also important for chronic autoinflammatory diseases such as rheumatoid arthritis, inflammatory bowel diseases including Chron’s disease and ulcerative colitis, multiple sclerosis, and psoriasis. While transforming growth factor-β and IL-6 are important for differentiation of the subset, IL-23 produced by dendritic cells activated by TNF-α is essential for its maintenance and activation. It is widely accepted that a vicious circle among TNF-α, IL-23, and IL-17 is established in psoriasis. Monoclonal antibodies against TNF-α, IL-12/23p40 subunit, IL-23p19, and IL-17 are clinically used and very effective [22,23,24,25].

IL-17 has multiple physiological functions. It activates macrophages to express TNF-α and IL-1β [26] and fibroblasts to produce IL-6, IL-8, and matrix metalloproteinases [27]. These cytokines are important for inflammatory process and tissue remodeling. It also stimulates blood endothelial cells to produce chemoattractants and express VCAM-1 and ICAM-1 in a p38 MAPK-dependent manner, which helps immune cells evade from blood to tissues [28]. Epithelial cells stimulated by IL-17 express IL-8 and granulocyte colony-stimulating factor, inducing migration and activation of neutrophils [29]. In addition, IL-17 was found to synergistically enhance TNF-α-induced production of these cytokines. IL-17 also increases production of antimicrobial peptides (AMPs), which are part of the innate immune response and are found among all classes of life [30,31]. These peptides kill bacteria, mycobacteria, enveloped viruses, and fungi. Skin-resident commensal microbes produce their own AMPs, act to enhance the normal production of AMPs by keratinocytes, and are beneficial to maintaining inflammatory homeostasis by suppressing excess cytokine release after minor epidermal injury [32]. IL-22, which is another cytokine expressed by Th17, enhances AMP production together with IL-17 [31]. This cytokine induces epidermal proliferation [33]. IL-26, an antimicrobial protein that has the ability to directly kill extracellular bacteria, is also secreted by Th17 [34]. IL-26 was more strongly expressed in lesions from the self-limited tuberculoid leprosy compared with expression in progressive lepromatous patients. IL-26 directly bound to *Mycobacterium leprae* in axenic culture and reduced bacteria viability. Thus, Th17 is important for neutrophilic infiltration, epidermal proliferation called acanthosis, and innate immune responses including enhanced production of AMPs, which are main histological findings characteristic to psoriasis.

## 3. Involvement of Th17-Related Cytokines in AD

### 3.1. Th17-Related Cytokine Expression in AD

Possible involvement of Th17 in the pathogenesis of AD has been reported by some researchers [16,17,18,19,20]. The percentage of IL-17-poducing CD4+ T cells in peripheral blood from AD patients was increased and associated with severity of AD [16]. There was a significant correlation between the percentages of IL-17+ and IFN-γ+ cells, although the percentage of Th17 cells was not closely related to Th1/Th2 balance. Immunohistochemically, IL-17+ cells infiltrated in the papillary dermis of AD lesional skin. There was another study showing that more IL-17-producing cells infiltrated into the site of an atopy patch test than in healthy skin [17]. IL-17 secretion was not enhanced by IL-23, IL-1β, or IL-6, but was enhanced by the *Staphylococcus aureus*-derived superantigen staphylococcal enterotoxin B. Another group, however, showed that the number of Th1 and Th17 subsets in peripheral blood from AD patients was significantly decreased, but that of the Th2 subset was similar to that of normal controls [35]. The frequency of Th17 cells showed a significant, negative correlation with serum thymus and activation-regulated chemokine (TARC) and IgE levels and the number of eosinophils. It was recently reported that the frequency of IL-13+ cells, which was correlated with IgE levels and SCORing AD (SCORAD) score, was associated with IL-22-producing T cells in severe AD patients [36]. When examining skin samples from AD, psoriasis, and healthy controls, IL-17 expression level was increased in AD skin compared to normal skin, although it was much lower than that in psoriasis skin [18]. By contrast, Th2 cells were significantly elevated in AD. Distinct IL-22-producing CD4+ and CD8+ T-cell populations were significantly increased in AD skin compared with psoriasis. IL-22-producing CD8+ T-cell frequency correlated with AD disease severity. Taken together, IL-22 rather than IL-17 is dominant in AD skin, while IL-17 is dominant in psoriasis skin.

It is a recent trend to divide various diseases into subgroups by cytokine production. Two subgroups of asthma, intrinsic and extrinsic, have been proposed [37]. Extrinsic asthma is triggered by allergens and is characterized by enhanced Th2 cytokine production. Intrinsic asthma is characterized by later onset in life, female predominance, higher degree of severity, and more frequent association to nasosinusal polyposis. This nonallergic asthma characterized by Th1 production can be objectively distinguished from allergic asthma based on negative skin tests to usual aeroallergens. Similar subgroups have been proposed for AD: intrinsic AD and extrinsic AD [38,39]. Extrinsic or allergic AD shows high total serum IgE levels and the presence of specific IgE for environmental and food allergens, whereas intrinsic or nonallergic AD exhibits normal total IgE values and the absence of specific IgE [38]. Extrinsic AD is much more common than intrinsic AD, which is approximately 20% with female predominance. The clinical features of intrinsic AD include relative late onset, milder severity, a higher frequency of metal allergy, and Dennie–Morgan folds, but no ichthyosis vulgaris or palmar hyperlinearity. The skin barrier dysfunction and filaggrin gene mutations are not a feature of intrinsic AD. The intrinsic type is characterized by the lower expression of IL-4, IL-5, and IL-13, and the higher expression of IFN-γ. In patients with intrinsic AD, Th17-related cytokines are highly expressed [39]. Positive correlations between Th17-related molecules and SCORAD scores were only found in patients with intrinsic AD, whereas only patients with extrinsic AD showed positive correlations between SCORAD scores and Th2 cytokine levels and negative correlations with keratinocyte differentiation markers such as loricrin and periplakin. Another classification of AD was proposed: Asian type and European American type. A principal component analysis using real-time PCR data clustered the Asian AD phenotype between the European American AD and psoriasis phenotypes [19]. Significantly higher Th17 and lower Th1 gene induction typified AD skin in Asian patients. Moreover, AD lesional skin in pediatric patients was reported to contain more Th17-related cytokines and AMPs than that in adult patients [20]. Taken together, it may be safely said that IL-17 expression in lesional skin of a certain group of AD patients is higher than that in normal skin. It is yet to be elucidated whether IL-17 plays a critical role in AD as it does in psoriasis.

With regards to serum levels of cytokines, it was reported that IL-22 in AD patients significantly correlated with disease activity, while IL-17 did not reflect disease burden [35].

IL-26 is another cytokine produced by Th17 cells. It was recently reported that IL-26 mRNA expression levels were elevated in AD lesional skin compared with healthy controls and that IL-26-producing cells were increased in AD lesional skin by immunohistochemistry [40]. IL-26 may play an important role for bridging between Th17 and Th2 responses, resulting in the development of AD.

### 3.2. AD Mouse Models and Th17-Related Cytokines

With regards to basic research, IL-17 was reported to regulate Th2 responses in the mouse AD model [41]. IL-17 triggered the production of IL-4 by Th2 cells. The lack of IL-17A reduced dermatitis and IL-4 production as well as IgE production. It is also known that IL-17 promotes the differentiation of B cells to IgE-producing plasma cells [42]. A murine epicutaneous infection model showed that *Malassezia*, which has been reported to exacerbate AD skin conditions, selectively induced IL-17-related cytokines [43]. This process is important for preventing fungal infection through the skin. Under skin barrier dysfunction mimicking AD, the presence of *Malassezia* aggravated cutaneous inflammation, which was dependent on IL-23 and IL-17. A CCR6+ Th17 subset of memory T cells specific to *Malassezia* was more frequently detected in AD skin than in healthy skin. Thus, the *Malassezia*-induced Th17 response is important not only in antifungal immunity but also in worsening skin inflammatory conditions. TNF-like weak inducer of apoptosis (TWEAK) and its receptor fibroblast growth factor (FGF) inducible 14 are highly expressed in AD skin lesions. BALB/c mice deficient in this molecule showed attenuated skin inflammation in an AD model, accompanied by less infiltration of inflammatory cells and lower local levels of proinflammatory cytokines, including TWEAK, TNF-α, and IL-17 [44]. Thus, IL-17 may be an important mediator of TWEAK-induced AD-like inflammation. The numbers of ILC3, expressing IL-17, in the skin of AD-induced mice were increased, and that neutralizing IL-17A delayed development of AD in the mouse model [45]. ILC3 induced IL-33 production by keratinocytes and fibroblasts. IL-26, another Th17 cytokine, promoted production of various cytokines such as IL-8, IL-1β, CCL20, IL-33, and β-defensin 2 by keratinocytes through phosphorylation of signal transducer and activator of transcription 1 and signal transducer and activator of transcription 3 [37]. JAK inhibitors, which are promising drugs for AD, blocked IL-26-induced cytokine production in keratinocytes. Injection of IL-26 exacerbated an oxazolone-induced AD mouse model and upregulated Th2 and Th17 cytokine expression in vivo. Thus, basic research findings also support the idea that IL-17 and its related cytokines are important in AD pathogenesis.

## 4. Expression of Th17 Cytokines in Other Skin Diseases

Some reports have suggested importance of IL-17 in various skin diseases other than psoriasis and AD. For example, IL-17 expression is reported to be increased in sera or lesional skin of systemic lupus erythematosus [46]. IL-17A expression was significantly increased in the involved skin and sera of systemic sclerosis patients, suggesting possible roles in the development of the disease [47,48]. Our group reported that lesional skin of mycosis fungoides (MF) and Sezary syndrome (SS), representative diseases of CTCL, contained a high amount of IL-17 and IL-22, the latter of which was dominant [49]. Serum IL-22 levels were significantly positively associated with disease activity in MF/SS. There are many clinical similarities between MF/SS and AD. Both diseases are characterized by persistent erythematous lesions with severe itch, sometimes developing to erythroderma. Bacterial, fungal, and viral infections such as impetigo contagiosum, phlegmone, tinea corporis, and herpes simplex are frequently accompanied. The number of eosinophils in peripheral blood is often increased and serum TARC level is elevated. Dominancy of IL-22 over IL-17 is seen in both diseases, which may explain clinical pictures of these diseases. Taken together, Th17 cytokine expression is widely seen in various skin diseases.

## 5. Clinical Trial for AD Targeting Th17 Cytokines

### 5.1. Clinical Effect of Ustekinumab on AD

Ustekinumab is a recombinant human IgG1 monoclonal antibody against p40, which is a common subunit of IL-12 and IL-23 [25]. This drug is used for psoriasis vulgaris and psoriatic arthritis. A previous report about four male patients with severe AD between 23 and 29 years old, whose skin eruption had been refractory to oral corticosteroids, phototherapy, and to at least two systemic drugs, showed improvement of SCORAD and itch score after treatment with 45 mg of ustekinumab injection at 0, 4 weeks, and every 12 weeks afterwards [50]. A 16-year-old woman with over a 13-year history of severe AD was also reported to show clearance of severe itch after one injection of ustekinumab 45 mg, although it had been resistant to phototherapy and systemic agents such as cyclosporine and azathioprine [51]. Complete remission was achieved with ustekinumab at the fourth week and every 12 weeks afterwards. There are, however, other reports denying efficacy of ustekinumab [52,53,54]. A 21-year-old man with a 7-year history of severe psoriasis refractory to conventional systemic treatments and childhood AD history was treated with ustekinumab [52]. The patient was without major respiratory symptoms and AD lesions within a period of five years before the date of starting the treatment. After an eight-month interval, ustekinumab was restarted and severe AD appeared on the neck and lower limbs. Peripheral blood eosinophilia and abnormal increase in serum IgE were also noted. The eczema lasted during 12 months of follow up after the final dose of ustekinumab, supporting the diagnosis of AD. Another group reported two psoriatic patients with high serum IgE levels, in whom ustekinumab completely improved psoriasis but paradoxically provoked or exacerbated AD-like symptoms [54]. Recently, the possibility of ustekinumab for AD was tested in clinical trials all over the world [55,56]. In the randomized, placebo-controlled, phase II study in Japan, 79 patients aged between 20 and 65 years with severe or very severe AD entered a 12-week, double-blind treatment period during which they received (1:1:1) ustekinumab 45 mg, 90 mg, or placebo subcutaneous injections at weeks 0 and 4, with follow-up until week 24 [55]. There was no significant improvement with ustekinumab treatment in least-squares mean change from baseline Eczema Area and Severity Index score at week 12. In another phase II, double-blind, placebo-controlled study, 33 patients with moderate-to-severe AD were randomly assigned to either ustekinumab (*n* = 16) or placebo (*n* = 17), with subsequent crossover at 16 weeks and last dose at 32 weeks [56]. Background therapy with mild topical steroids was allowed. Study endpoints included clinical and biopsy-based measures of tissue structure and inflammation, using protein and gene expression studies. The ustekinumab group achieved higher clinical responses at 12, 16 (the primary endpoint), and 20 weeks compared to placebo, but the difference between groups was not significant. The AD molecular profile/transcriptome showed early robust gene modulation, with sustained further improvements until 32 weeks in the initial ustekinumab group. It is still possible that a certain subset of patients with AD may particularly benefit from ustekinumab. However, it may be safely said that ustekinumab is much more effective in psoriasis than in AD.

### 5.2. Other Biologics and AD

There have been anecdotal case reports showing exacerbation of AD-like dermatitis after treatment with anti-IL17 antibody for psoriasis [57,58,59]. A 31-year-old man with severe psoriasis treated with ixekizumab presented with a one-month history of pruritic lesions [57]. The patient had no personal or family history of allergy, AD, or eczema. Treatment with ixekizumab, started 14 months before the episode, achieved an almost complete response of his skin disease. The patient developed the palms exudative erythematous plaques on the hands, the trunk, and the medial aspects of the limbs. Histologically, epidermal spongiosis with vesicle formation with dermal edema and lymphocytic perivascular infiltrate with some eosinophils were seen. Ixekizumab may suppress the expression of keratinocyte-derived AMPs, increasing the risk of bacterial and fungal infection, which can lead to the eczematous phenotype. Another group reported a 70-year-old woman with intractable psoriasis vulgaris treated with 300 mg of secukinumab [58]. Six months after the first dose of secukinumab was administered, psoriasis lesions were cleared. Blepharitis, cheilitis, and dermatitis around the nose, however, developed four days after every secukinumab injection. A common skin-resident fungus *Malassezia* may account for these skin lesions, which might be more adequately named seborrheic dermatitis-like lesions instead of AD-like eczema. Fezakinumab, an anti-IL-22 antibody, has been recently tried for treatment of AD. A randomized, double-blind, placebo-controlled trial with intravenous fezakinumab monotherapy every 2 weeks for 10 weeks, with follow-up assessments until 20 weeks, was performed for adult patients with moderate-to-severe AD [60]. At 12 weeks, the mean declines in SCORAD for the entire study population were not different between the fezakinumab arm and the placebo arm. In the severe AD patient subset, SCORAD decline was significantly stronger in the drug-treated patients than placebo-treated patients at 12 weeks. Common adverse events were upper respiratory tract infections. Although this drug significantly downregulated gene expression of multiple immune pathways including Th1, Th2, and Th17, it was clinically effective only in patients with severe AD expressing a high level of IL-22 at the baseline [61]. Taken together, biologics targeting Th17-related cytokines are not promising for the treatment of AD, except for anti-IL-22 antibody.

## 6. Th2 Cytokines Are Key Players in AD

If antibodies against Th17 cytokines are not useful for AD, what is the best therapeutic target? Th2-type immune responses induced by allergens such as mites and pollen are associated with worsening of AD skin conditions. Th2 immune responses including increased IL-4/IL-13 expression induce the production of IgE and TARC [62]. Serum TARC levels are useful as a marker of the disease activity of AD, which are more sensitive than serum IgE levels, lactate dehydrogenaselevels, and peripheral blood eosinophil counts [63]. Recently, thymic stromal lymphopoietin (TSLP) and periostin have been discovered to be key cytokines in Th2-dominant microenvironment [64,65]. TSLP-activated dendritic cells primed naïve T cells to produce IL-4, IL-5, IL-13, and TNF-α, while down-regulating IL-10 and IFN-γ [64]. TSLP was highly expressed by epithelial cells, especially keratinocytes from AD patients. Th2 cytokines IL-4 and IL-13 stimulated fibroblasts to produce periostin, which interacted with a functional periostin receptor on keratinocytes [65]. It induces production of proinflammatory cytokines, which consequently accelerated Th2-type immune responses. It was recently reported that the TSLP receptor was expressed by some T cell subsets from patients with Th2-dominant diseases and that TSLP directly enhanced IL-4/IL-13 production by T cells [66,67]. Taken together, these Th2-related cytokines, IL-4/IL-13, TSLP, and periostin are driving factors of a vicious circle in AD and can be good candidates for therapeutic targets. Indeed, dupilumab, a monoclonal antibody against the IL-4 receptor, is very effective for severe AD [68].

## 7. The Importance of Th17 Cytokines in the Skin

Skin is a border between the outer world and the body. It is important to block invasion of foreign materials such as poison and parasites. Trauma and loss of skin barrier function in diseased skin, such as in AD and psoriasis, causes a high risk of the invasion. In both conditions, mechanical stimulation can be the cause of onset. It is called the Kobner phenomenon in psoriasis. High proliferation of epidermal cells and rapid keratinization are seen in both skin conditions. Expression of K6 and K16 is increased and that of K1 and K10 is decreased [69,70]. Increased expression of cathepsin D, which modulates expression of involucrin, loricrin, and filaggrin, has been reported [71].

Cytokine expression and cellular infiltration in psoriasis lesional skin is also very similar to what is seen in cases of trauma. IL-17A, which is expressed both in wounded skin and psoriasis skin, plays important roles in wound healing, tissue regeneration, and carcinogenesis [72]. The cytokine mediates activation of EGFR, which is critical for the expansion and migration of Lrig1+ stem cells, promoting wound healing [73]. FGF2 or IL-17 deficiency resulted in impaired epithelial proliferation, increased pro-inflammatory microbiota outgrowth, and consequently worse pathology in a mouse colitis model [74]. The dysregulated microbiota in the model induced transforming growth factor-β1 expression, which in turn induced FGF2 expression mainly in regulatory T cells. Thus, microbiota-driven FGF2 and IL-17 cooperate to repair the damaged intestinal epithelium. Injury of the mucosa causes fast expansion of Th17 cells and their induction [75]. Th17 cells produce various cytokines, such as TNF-α, IL-17, and IL-22, which can promote cell survival and proliferation, and thus tissue regeneration, in several organs including the skin [72]. IL-17 promoted macrophage infiltration into wounded skin and scar formation through an indirect mechanism [76]. Depletion of macrophages with clodronate liposomes abrogated the effect of IL-17. Levels of monocyte chemotactic protein (MCP) 1, MCP2, and MCP3 were increased by IL-17 stimulation. IL-17 induced the infiltration of a specific subtype of macrophages to aggravate fibrosis through an MCP-dependent mechanism. On the other hand, Th17 cells are potentially pathogenic if not tightly controlled. Failure of these control mechanisms can result in chronic inflammatory conditions such as psoriasis.

Infiltration of neutrophils and macrophages is commonly seen, and AMP expression is increased both in wounded skin and in psoriasis skin. Vascular formation is prominent in the dermis. Taken together, it may be safely said that psoriasis is a hyperreaction to traumatic stimuli with the background of genetics, circumstances, and metabolic status (Figure 1).

## 8. Ideal Biologics for Psoriasis, AD, and CTCL

Clinical effects of biologics are remarkable, and many diseases including skin diseases are currently treated with them. The cost of developing these agents, however, is huge and it is important to predict which molecule is promising or not. First of all, the molecule should be highly expressed in lesional skin compared to normal skin. Second, the expression of the target should be highest in the disease among various skin diseases. It is important not to block bystander cytokines. IL-22 is highly expressed in psoriasis, but its expression is lower in AD. IL-17 expression in AD is not as high as that in psoriasis. Anti-IL-22 antibody is not effective in psoriasis, and anti-IL12/23p40 or anti-IL-17A antibodies are not promising as treatment for AD. Only the anti-IL-22 antibody may be beneficial for a certain group of AD patients (Table 1). Blocking cytokines uniquely expressed in the disease is important to stop a vicious circle. Third, it would be safer to block interactions between cytokines and their receptors than targeting surface molecules. Exclusion of a certain cell type by antibody-dependent cellular cytotoxicity induced by biologics may cause unexpected severe effects. This kind of strategy targeting surface molecules should be adopted for the treatment of malignancy [79,80].

## 9. Conclusions

It is true that Th17 cells infiltrate into lesional skin of AD, and Th17-related cytokines are expressed there. These cytokines have some roles in the inflammatory process of the disease. It is, however, also true that IL-17 in AD is not as important as it is in psoriasis. Expression of IL-17 and other Th17-related cytokines is widely seen in many skin diseases. My current assumption is that IL-17 expression in the skin is a defense mechanism against foreign materials invading the body.

## Figures and Tables

**Figure 1 ijms-21-01314-f001:**
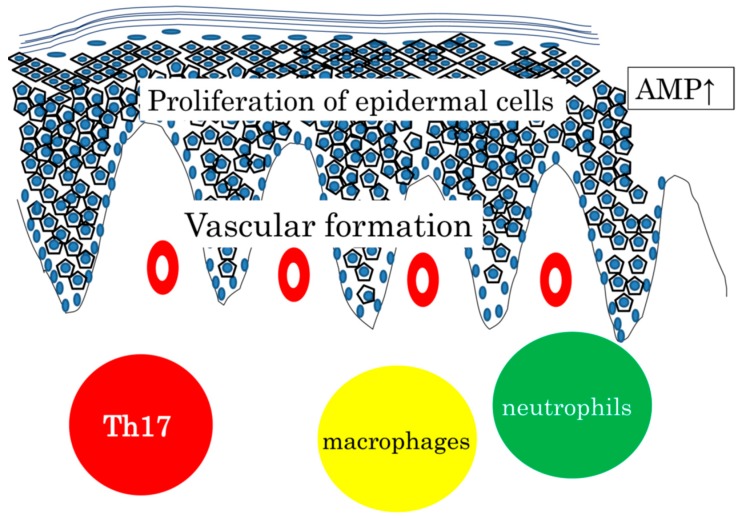
Proliferation of epidermal cells, AMP expression, vascular formation, and infiltration of Th17, neutrophils, and macrophages, commonly seen in wounded skin and in psoriasis skin. We sometimes experience fungal infection in patients treated with anit-IL7A or anti-IL17 receptor A antibody. Considering that Th17 cytokines are expressed in various skin diseases with barrier dysfunction and that antibodies against IL-12/IL-23p40 or IL-17A are not effective for AD, IL-17 and its related cytokine expression can be regarded as a common defense mechanism in the skin. Relatively low AMP expression leads to a higher risk of infection seen in AD and CTCL [77,78]. Targeting the Th17 cytokine has been successful for psoriasis, but may not be always the case in other skin diseases.

**Table 1 ijms-21-01314-t001:** Antibodies against Th17 and Th2-related cytokines for atopic dermatitis (AD).

Cytokines	Clinical Response	References
IL-17A	Worsening of dermatitis	[57,58,59]
IL-12/23p40	No effect in clinical trial	[55,56]
IL-22	Effective for some patients	[60,61]
IL-4/13	Very effective	[68]

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
