# Peer review of "The Role of Th17-Related Cytokines in Atopic Dermatitis"

_ijms, 2020, doi:10.3390/ijms21041314_

Round 1
Reviewer 1 Report
see attached file

Author Response
Comment 1:
The organization of the review needs some improvement. For instance, the section 3.2 (AD subgroups and cytokine expression) maybe switched with the section 3.1. Also, the title of section 3 (Expression of Th17-related cytokines in AD) doesn’t seem to cover all of the information discussed in 3.2.
Response: Thank you very much for your suggestion. I agree that some contents of the section 3.2 can move to the section 3.1. I have combined these two sections and switched some sentences. I also have changed the title of section 3 as “Involvement of Th17-related cytokines in AD.”
Comment 2:
The author needs to use consistent definition for IL-22-expressing T cells. In some cases, he refers IL-22 as one of Th17 cytokines, but in other cases, he used the name of Th22. Although both definitions might be interchangeably used in the scientific community, it is important for the author to use a single definition in one review.
Response: Thank you very much for your advice. IL-22 is one of Th17 cytokines and I have deleted the name of Th22.
Comment 3:
I did not find the Figure 1/2 are informative. I would like to see one figure or a table to compare and contrast the data collected from different studies or clinical trials. The goal is to help readers quickly understand the efficacy of targeting Th17-related cytokines in psoriasis and AD patients.
Response: According to your advice, I have deleted figures 1 and 2. I have made a new Table 1 summarizing the data collected from different studies or clinical trials about biologics and AD.
Comment 4:
The author may need to modify the title as “The role of Th17-related cytokines in atopic dermatitis and Psoriasis”
Response: Thank you very much for the useful comment. I have changed the title as “The Role of Th17-Related Cytokines in Atopic Dermatitis” because the manuscript covers mainly the role of IL-17-related cytokines in atopic dermatitis.
Comment 5:
All of the Greek letters are missing from the PDF.
Response: I am very sorry for that. I have corrected them.
Reviewer 2 Report
The manuscript covers very important and interesting issue. Atopic dermatitis is an inflammatory disease of the skin which may occur at any age.
The manuscript is an exhaustive review paper on pathogenesis of atopic dermatitis and the role of IL-17 in this process. Th17 cells infiltrate into lesional skin of AD and Th17-related cytokines are expressed there playing important role in inflammatory process.
Detailed and careful elaboration of the text should be emphasized. The work is also interesting from practical point of view, especially in the aspect of atopic dermatitis treatment with biologics.
The methods, field, design of paper are actual and fit to the journal. The authors cited current literature.
I would suggest to change the title because the manuscript covers mainly the role of IL-17 in atopic dermatitis, which the author, himself admit: “I would like to focus on importance of Th17-related cytokines in pathogenesis and treatment of AD”. It was inevitable that the author compare some aspects with psoriasis because of evident role of IL-17 in its pathogenesis.
Author Response
Comment 1: I would suggest to change the title because the manuscript covers mainly the role of IL-17 in atopic dermatitis, which the author, himself admit: “I would like to focus on importance of Th17-related cytokines in pathogenesis and treatment of AD”. It was inevitable that the author compare some aspects with psoriasis because of evident role of IL-17 in its pathogenesis.
Response: Thank you very much for the useful comment. I have changed the title as “The Role of Th17-Related Cytokines in Atopic Dermatitis” because the manuscript covers mainly the role of IL-17-related cytokines in atopic dermatitis.